# Modelling the ultra-strongly coupled spin-boson model with unphysical modes

Neill Lambert[1], Shahnawaz Ahmed [1,2], Mauro Cirio[3] & Franco Nori [1,4]

A quantum system weakly coupled to a zero-temperature environment will relax, via spontaneous emission, to its ground-state. However, when the coupling to the environment is ultra-strong the ground-state is expected to become dressed with virtual excitations. This regime is difficult to capture with some traditional methods because of the explosion in the number of Matsubara frequencies, i.e., exponential terms in the free-bath correlation function. To access this regime we generalize both the hierarchical equations of motion and pseudomode methods, taking into account this explosion using only a biexponential fitting function. We compare these methods to the reaction coordinate mapping, which helps show how these sometimes neglected Matsubara terms are important to regulate detailed balance and prevent the unphysical emission of virtual excitations. For the pseudomode method, we present a general proof of validity for the use of superficially unphysical Matsubara-modes, which mirror the mathematical essence of the Matsubara frequencies.

[1] Theoretical Quantum Physics Laboratory, RIKEN Cluster for Pioneering Research, Wako-shi, Saitama 351-0198, Japan. [2] Wallenberg Centre for Quantum Technology, Department of Microtechnology and Nanoscience, Chalmers University of Technology, 412 96 Gothenburg, Sweden. [3] Graduate School of China Academy of Engineering Physics, Haidian District, Beijing 100193, China. [4] Department of Physics, University of Michigan, Ann Arbor, MI 48109-1040, USA. Correspondence and requests for materials should be addressed to N.L. (email: nwlambert@gmail.com) or to M.C. (email: cirio.mauro@gmail.com)

The spin-boson model is a cornerstone of the theory of open-quantum systems, and its elegance often belies its power to describe a wide range of phenomena[1–3]. It not only allows us to understand the relationship between quantum dissipation and classical friction, but is a powerful model to study topics ranging from physical chemistry to quantum information. Practically speaking, a number of perturbative approaches and assumptions such as the Born–Markov and rotating-wave approximation (RWA) are usually employed to obtain tractable solutions[2]. However, research areas, such as energy transport in photosynthetic systems[4–9], quantum thermodynamics[10,11], and the ultra-strong coupling regime in artificial light-matter systems[12–17], have demanded the development of numerically exact methods to explore non-perturbative and non-Markovian parameter regimes[18–20], which are out of reach of traditional approaches.

In the limit of a discrete environment consisting of a single bosonic mode, as arises in cavity QED (cQED)[21], the non-perturbative limit, in which the coupling is a significant fraction of the cavity frequency, is sometimes referred to as the ultra-strong coupling (USC) regime[16,17]. This regime harbors a range of new physics, including higher-order coupling effects, the possibility to excite two atoms with one photon[22], the ability to prepare Bell and GHZ states in cQED[23], and the potential to generate a ground state which contains virtual excitations[12,24–26]. In the latter, the excitations are called virtual because they are energetically trapped in the hybridized light-matter ground state. A correct theoretical understanding of this trapping, such that unphysical emission from the ground state is avoided, was only developed recently[27]. It is now understood that nonadiabatic external forces must be applied to transmute them into real, observable, excitations[24,28–31].

Numerical simulations[32–34] have suggested that a similar phenomenon occurs when a spin, or two-level system, is ultra-strongly coupled to a continuum environment (i.e., an infinite number of bosonic modes), as traditionally described with the spin-boson model[25]. This scenario is now becoming experimentally accessible in one-dimensional transmission lines[19,35], and superconducting metamaterials[36–38]. In addition, even at nonzero temperatures, it has been shown that virtual excitations can influence the efficiency of a photosynthetic quantum heat engine[39].

Methods and techniques are needed which are both quantitatively accurate and able to provide a qualitative understanding of the physics in this non-perturbative low-temperature regime. However, many successful numerical methods used to study the spin-boson model away from the normal perturbative limits, such as the hierarchical equations of motion (HEOM) technique[40,41] and the pseudomode method[42,43], are limited in their ability to access low temperatures.

Here, we go beyond these restrictions and show how generalized versions of the HEOM and pseudomode methods can help us understand the nature of the ground state in the continuum, and explain how virtual excitations are trapped therein. In particular, in our generalization of the pseudomode method, the original continuum environment is replaced by a discrete set of modes[42] which not only help to quantitatively describe the correct low-energy non-perturbative physics but also help simplify the model to the essential mathematical elements needed to give a physical intuition about properties of the ground state. In addition, the methods we develop herein may also enable the exploration of virtual processes in quantum field theory, which are usually considered not physically accessible[44]. They can also assist in the exploration of new physics and the development of applications in coupled light-matter systems[16,17], and allow the modeling of complex light-harvesting processes in new parameter regimes[9].

We begin with an overview of our main results. We then introduce the spin-boson model and free-bath correlation functions, and provide an intuitive explanation of why omitting the apparently negligible Matsubara terms can have large consequences, even in the weak coupling regime. Next we demonstrate our correlation function fitting method for the HEOM, before turning to the pseudomode method and the reaction coordinate mapping to more transparently explain what happens when Matsubara terms are ignored in the ultra-strong coupling regime. Finally, we compare all three methods, with and without Matsubara contributions, and show their predictions for the dynamics and steady-state occupation of certain environment modes.

## Results

**Overview.** We now summarize our main results in detail. As mentioned, using the HEOM technique in the low-temperature limit is difficult[45–47], typically making this regime inaccessible. This is because the HEOM relies on a decomposition of the bath correlation function into a sum of exponentials. Unfortunately, due to the physical constraint disallowing Hamiltonians unbound from below (i.e., that the environment only consists of positive frequency modes), even a simple Lorentzian spectral density gives correlation functions which cannot be analytically decomposed into a finite sum. The same restriction has historically applied to the pseudomode method[42,43], as we will describe below.

To overcome this difficulty, we separate the correlation function into an analytical part, comprised of a finite number of exponentials, and the Matsubara part, given by an infinite sum of exponentials (the latter of which was neglected in other works studying the zero-temperature limit of the HEOM method[48,49]). In the zero-temperature limit, we analytically integrate the infinite sum, and then fit it with a biexponential function. Fitting the total correlation function to exponentials for use with the HEOM has also been explored in refs. [45–47,50], but our approach allows us to limit the fitting error[51] to the Matsubara component, and gives us physical insight into the role of the different contributions to the correlation function. The fitting inevitably introduces some error in the system dynamics, which we analyze in detail in Supplementary Notes 3 and 4.

By comparing results with and without this Matsubara contribution, we find that the neglect of the Matsubara terms in both the HEOM formulation, and a generalized pseudomode method, induces a very specific error in the dynamics and steady state. This error corresponds to an unphysical system temperature, even at weak coupling, due to violation of detailed balance. Conversely, when including the Matsubara terms, detailed balance is restored, albeit with a finite error due to the fit. In the ultra-strong coupling regime, we find that, via comparison with the reaction coordinate method[10,52–54] neglecting the Matsubara terms leads to an unphysical emission of photons from the ground state of the coupled light-matter system (to which we will refer as our main example).

In generalizing the pseudomode method, which employs the fit of the Matsubara frequencies in the form of two additional zero-frequency Matsubara modes with non-Hermitian coupling to the system, we find that it can exactly reproduce the full HEOM results for all parameter regimes. It can also be used to give meaning to the auxiliary density operators (ADOs) of the HEOM, indicating a strong relationship between the two approaches. To account for the unusual form of the Matsubara modes, we explicitly generalize the proof of validity of the pseudomode method[42,43]. Our derivation shows that by combining the non-Hermitian Hamiltonian together with what we call a pseudo-Schrödinger equation, the Dyson equation for the reduced

dynamics of the system is formally equivalent to one where the system is physically interacting with the original continuum environment.

**The spin-boson model.** The iconic spin-boson model considers a two-level system (the spin, or qubit) in a bath of harmonic oscillators with the total system-bath Hamiltonian given by (setting $\hbar = 1$ throughout):

$$H = \frac{\omega_{\mathrm{q}}}{2}\sigma_z + \frac{\Delta}{2}\sigma_x + \sum_k \omega_k b_k^\dagger b_k + \sigma_z \tilde{X} \;, \quad (1)$$

where $\omega_{\mathrm{q}}$ is the qubit splitting, $\omega_k$ is the frequency of the $k$th bath mode, $\Delta$ is the tunneling matrix element, $\sigma_{z(x)}$ are the Pauli matrices acting on the qubit. For later use, we define $\bar{\omega} = (\omega_{\mathrm{q}}^2 + \Delta^2)^{1/2}/2$, as the free qubit eigenfrequency. The $k$th mode of the bath, associated with annihilation operators $b_k$, interacts with the qubit via the operators $\tilde{X}_k = g_k/\sqrt{2\omega_k}(b_k + b_k^\dagger)$ in terms of the couplings $g_k$, so that $\tilde{X} = \sum_k \tilde{X}_k$.

The effect of the bath can be considerably simplified when the initial state of the environmental modes is Gaussian, and in a product state with the system (the qubit). Specifically, we assume the bath to be in a thermal state at a temperature $T$. In this case, the influence of the environment is contained in the two-time correlation function $C(t) = \langle \tilde{X}(t)\tilde{X}(0)\rangle$. The correlation function of the free bath, when it is not in contact with the system, can be written (in the continuum limit) as,

$$C(t) = \int_0^\infty \mathrm{d}\omega \frac{J(\omega)}{\pi}\left[\coth\left(\frac{\beta\omega}{2}\right)\cos(\omega t) - \mathrm{i}\sin(\omega t)\right] \;. \quad (2)$$

Here, $J(\omega) = \pi \sum_k g_k^2/2\omega_k \delta(\omega - \omega_k)$ is the spectral density, which parameterizes the coupling coefficients $g_k$, and $\beta = 1/k_{\mathrm{B}}T$ is the inverse temperature. Throughout this article, we focus on the following under-damped Brownian motion spectral density,

$$J(\omega) = \frac{\gamma\lambda^2\omega}{(\omega^2 - \omega_0^2)^2 + \gamma^2\omega^2} \;, \quad (3)$$

which is characterized by a resonance frequency $\omega_0$, a width $\gamma$, and a strength $\lambda$. A spectral density of this form is a convenient basis, in which one can represent a range of other spectral densities[55,56].

In the under-damped limit ($\gamma < 2\omega_0$), it is convenient to decompose the correlation function, for Eq. (3) in Eq. (2), as $C(t) = C_0(t) + M(t)$, where

$$C_0(t) = \frac{\lambda^2 e^{-\gamma t/2}}{4\Omega}\left[C_0^{\mathrm{R}}(t) + C_0^{\mathrm{I}}(t)\right] \;, \quad (4)$$

in terms of $C_0^{\mathrm{R}} = \coth[\beta(\Omega + i\Gamma)/2]\exp(i\Omega t) + \mathrm{H.c.}$ (where H.c. denotes Hermitian conjugation) and $C_0^{\mathrm{I}} = e^{-i\Omega t} - e^{i\Omega t}$, and

$$M(t) = -\frac{2\lambda^2\gamma}{\beta}\sum_{k>0}^\infty \frac{\epsilon_k e^{-\epsilon_k t}}{\left[(\Omega + i\Gamma)^2 + \epsilon_k^2\right]\left[(\Omega - i\Gamma)^2 + \epsilon_k^2\right]} \;, \quad (5)$$

with the definitions $\Gamma = \gamma/2$, $\Omega^2 = \omega_0^2 - \Gamma^2$, and $\epsilon_k = 2\pi k/\beta$ ($k \in \mathbb{N}$) for the Matsubara frequencies.

Intuitively, the $C_0(t)$ part of the correlation function characterizes the resonant part of the bath, with a shifted resonant frequency $\Omega$ and decay rate $\gamma/2$. On the other hand, the $M(t)$ part of the correlation function seems to have a less transparent description: it has no resonances but infinite sub-contributions which decay at rates equal to $\epsilon_k$ (hence we will name it the Matsubara correlation). One way to explore its meaning is to study what happens to the qubit dynamics after imposing $C(t) \rightarrow C_0(t)$, i.e., completely neglecting it. Note that this will induce an error even at zero temperature ($\beta \rightarrow \infty$) due to the competition

between the factor $\beta^{-1}$ and the Matsubara frequencies approaching the continuum.

To proceed with our intuitive analysis, it is worth considering the Fourier transform of the correlation function, i.e., the power spectrum $S(\omega) = \int_{-\infty}^\infty \mathrm{d}t\, C(t)e^{i\omega t} = J(\omega)[1 + \coth(\beta\omega/2)]$. From this expression, it is possible to check that the power-spectrum encodes the symmetry condition

$$S(\omega) = \exp(\beta\omega)S(-\omega) \;. \quad (6)$$

When the coupling to the environmental degrees of freedom is small compared with the qubit eigenfrequency $\bar{\omega} = (\omega_{\mathrm{q}}^2 + \Delta^2)^{1/2}/2$, the effect of the bath can be studied perturbatively (for example by using the Fermi golden rule). In this case, the qubit will absorb (relax) energy from (into) the environment at rates proportional to $S(-\bar{\omega})$ ($S(\bar{\omega})$) so that Eq. (6) encodes the physical meaning of the detailed balance condition. As a consequence, by neglecting the Matsubara correlations, we are then going to break this balance[57–59]. Nevertheless, the qubit will still reach an equilibrium thermal state at the effective temperature

$$\beta_{\mathrm{eff}} = \frac{1}{\bar{\omega}}\log\frac{S_0(\bar{\omega})}{S_0(-\bar{\omega})} \;, \quad (7)$$

where $S_0(\omega) = \int_{-\infty}^\infty \mathrm{d}t\, C_0(t)e^{i\omega t}$. The relation between $\beta_{\mathrm{eff}}$ and the actual temperature $\beta$ intuitively quantifies the effect of the Matsubara correlations when the coupling to the environment is very weak.

On the other hand, when the coupling with the environment starts to be a significant fraction of the system eigenfrequency, hybridization effects between the system and the bath become relevant. As it will be shown in a later section, the Matsubara correlations are essential to be able to correctly model both the non-Markovian and the equilibrium properties in this parameter regime (and which, in this case, were encoded in the detailed balance condition). We first describe the HEOM, and how the Matsubara term can be included, even at zero temperature, with a fitting approach.

**The hierarchical equations of motion.** The HEOM method can in principle describe the exact behavior of the system in contact with a bosonic environment, without approximations. The derivation can be found in refs. [40,41,48], and the general procedure can be described as follows. Using the Gaussian properties of the free bath, one can write down a formally exact time-ordered integral for the reduced state of the system (or equivalently, a path-integral representation). This is difficult to solve directly. However, by assuming that the free bath correlation functions can be written as a sum of exponentials, one can take repeated time derivatives to construct an exact series of coupled equations describing the physical density matrix, and auxiliary ones encoding the correlations between system and environment. These can be truncated at a level that gives convergent results.

The problem then lies in parameterizing the correlation functions of a given physical bath with a sum of exponentials. In practice, one can either fit[46,47,50] the correlation functions directly with exponentials or fit the spectral density using a sum of overdamped (Drude-Lorentz) or under-damped Brownian motion spectral densities[7,55,56]. However, for the latter, as one might expect from the discussion so far, the Matsubara frequencies in Eq. (5) become increasingly important at low temperatures. These frequencies, in the HEOM, are numerically challenging to take into account due to the increasing number of auxiliary density operators[45,60] (though using an alternative Padé decomposition with the HEOM has been explored as a way to optimally capture the influence of these terms[61]).

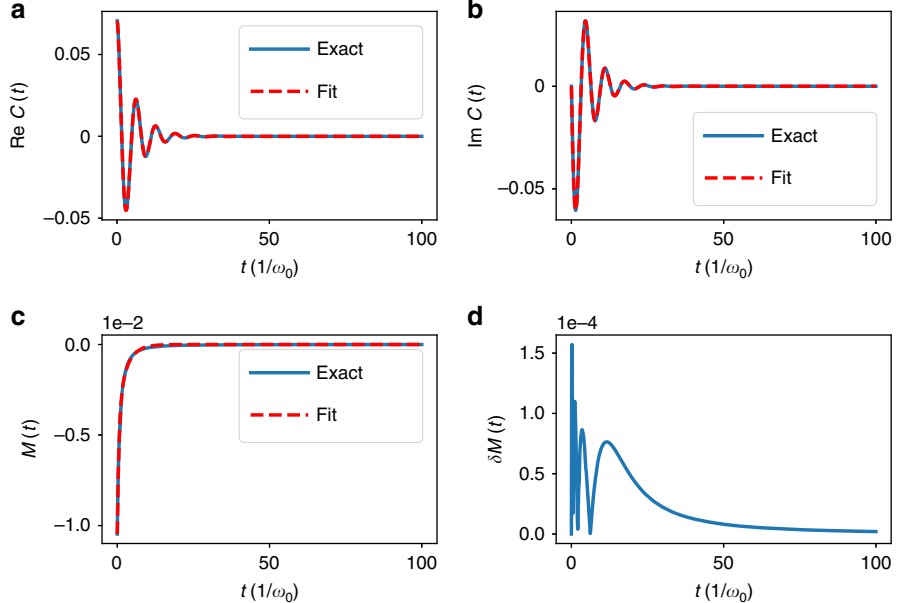

**Fig. 1** Free-bath correlation functions. The top two panels show (**a**) real and (**b**) imaginary parts of the correlation function for the under-damped Brownian motion spectral density with $\lambda = 0.4\omega_0, \gamma = 0.4\omega_0, T = 0$. The blue solid curves show the evaluation of the formula from Eq. (2) using Eqs. (4) and (8). The red-dashed curves show the reconstruction of the same using Eq. (9) to fit the Matsubara term, Eq. (8). In the bottom left panel (**c**), we explicitly plot the Matsubara part of the correlation function Eq. (4) alone, and its fit Eq. (9). The error in the fit is shown in the bottom right panel (**d**), which is also the same as the error in the real part of the correlation function. The imaginary part is exact, and has no error after the reconstruction

In the zero-temperature ($\beta \rightarrow \infty$) limit, the Matsubara frequencies $\epsilon_k = 2\pi k/\beta$ approach a continuum, i.e., $2\pi/\beta \rightarrow dx \rightarrow 0$ for $2\pi k/\beta \rightarrow x$. As a consequence, we can represent the Matsubara correlation in Eq. (5) as the integral

$$M(t) = -\frac{\gamma\lambda^2}{\pi} \int_0^\infty dx \frac{xe^{-xt}}{\left[(\Omega + i\Gamma)^2 + x^2\right]\left[(\Omega - i\Gamma)^2 + x^2\right]} \quad . \quad (8)$$

However, this integral representation does not give a direct solution in exponential form. Using a fitting procedure, we have found that we can capture the influence of these terms with a biexponential function,

$$M_{\text{biexp}}(t) = c_1 e^{-\mu_1 t} + c_2 e^{-\mu_2 t} \quad (9)$$

where $c_m$ and $\mu_i$ are real (for the choice of Matsubara decomposition we use here). Adding more exponential terms increases the accuracy of the fit only marginally for the parameter ranges we study here. In addition, each exponent leads to a large numerical overhead with the HEOM method, thus one would like to keep the number of exponents to a minimum. In Fig. 1, we give an example of the fitting of the correlation function.

Given the above decomposition, we can finally write the full equations of motion. However, a fully generic formulation of the HEOM[46] treats the real and imaginary parts of the correlation function separately, which turns the (for $\beta = \infty$) single non-Matsubara exponent in Eq. (4) into four exponents. It is more numerically convenient to reduce these to two exponents, following ref. [48], by defining (again, only for $\beta = \infty$ for notational simplicity) the new parameters $c_3 = \lambda^2 (1 - i)/4\Omega$, $c_4 = \lambda^2 (1 + i)/4\Omega$, $\mu_3 = -i\Omega + \Gamma$, and $\mu_4 = i\Omega + \Gamma$. Meanwhile, as described above, the Matsubara terms are entirely real, and given by Eq. (9).

In the HEOM itself, we denote the physical and auxiliary density matrices as $\rho_{\bar{n}}$ where $\bar{n} = [n_1, n_2, .., n_K]$, (where here $K = 4$), is a multi-index composed of non-negative integers $n_k$. The physical density matrix of the system, traced over the environment, is given by $\rho_{\bar{0}} = \rho_{[0,0,...,0]} \equiv \text{Tr}_E(\rho_T)$. Any other index denotes an auxiliary density operator which encodes the

correlations between system and environment, as we will discuss later. We use $\rho_{\bar{n}_{k\pm}}$ to denote a higher-order ADO, which differs from $\rho_{\bar{n}}$ in the $k$th index by ±1. For instance, $\rho_{0_{2+}} = \rho_{[0,1,0,...,0]}$. The equations of motion given by HEOM can be compactly written as

$$\dot{\rho}_{\bar{n}} = \left(-i\mathcal{L} - \sum_{k=1}^K n_k \mu_k\right)\rho_{\bar{n}} - i\sum_{k=1}^K \left(\mathcal{L}_k^- \rho_{\bar{n}_{k-}} + \mathcal{L}_k^+ \rho_{\bar{n}_{k+}}\right) \quad (10)$$

where $\mathcal{L}\rho = [H_s, \rho]$ and the $\mathcal{L}_k^\pm$ are Liouville space operators, depending on the spin-bath coupling operator and the exponential decomposition of the correlation function[40,41] given by $\mathcal{L}_k^- \rho_{\bar{n}_{k-}} = n_k(c_k^R[Q, \rho_{\bar{n}_{k-}}] + c_k^I\{Q, \rho_{\bar{n}_{k-}}\})$ and $\mathcal{L}_k^+ \rho_{\bar{n}_{k+}} = [Q, \rho_{n_{k+}}]$. Note again that this is not a generic construction[46], but is specific for the choice of decomposition of correlation functions we use here.

**Environment as a discrete set of modes**. Before discussing results predicted by the HEOM, it is useful to consider two complementary methods based on discrete decompositions of the environment. The idea that the behavior of an infinite continuum environment can be described by a finite set of discrete modes arises in both the methodology of pseudomodes[42,43,62–64] and the so-called reaction coordinate mapping[52–54]. The former is based on the identification of frequencies in the correlation functions which are then assigned to a set of unphysical pseudomodes[42,43]. In contrast, the reaction coordinate (RC) method is instead based on a formal mapping of the full Hamiltonian environment Hamiltonian to a single reaction coordinate and a residual (perturbative) environment.

**Pseudomodes model**. As shown in the seminal work of Garraway[42] (and recently confirmed and generalized in ref. [43]), as long as the free correlation function of a discrete set of modes accurately reproduces the correlation function of the full bath, their effect on a given system should be identical, a concept that recalls

in spirit Baudrillard: "The simulacrum is never that which conceals the truth—it is the truth which conceals that there is none. The simulacrum is true."[65].

From the discussion so far, and the generalized proof in ref. [43], it is evident that we can capture the full correlation function of the free environment, Eq. (2), with a single under-damped mode for the non-Matsubara part Eq. (4), and two additional modes, from the fitting procedure Eq. (9), which capture the Matsubara frequency contributions Eq. (5). By construction, at zero temperature, the resulting dynamics of the system coupled to these effective modes should obey the total Hamiltonian,

$$H_{pm} = \frac{\omega_q}{2}\sigma_z + \frac{\Delta}{2}\sigma_x + \sigma_z \sum_{i=1}^{3}\lambda_i(a_i + a_i^\dagger) + \sum_{i=1}^{3}\zeta_i a_i^\dagger a_i \ . \quad (11)$$

Here, $\zeta_1 = \Omega$, $\Omega = \sqrt{\omega_0^2 - \Gamma^2}$, $\zeta_2 = \zeta_3 = 0$, $\lambda_1 = \lambda/\sqrt{2\Omega}$, $\lambda_2 = \sqrt{c_1}$, $\lambda_3 = \sqrt{c_2}$ (where $c_1$ and $c_2$ are the coefficients of the fitted Matsubara terms in Eq. (9), and $\zeta_2 = \zeta_3 = 0$ because Eq. (9) contains no oscillating components).

The damping of each pseudomode is simply described by a Lindbladian with the corresponding loss rate,

$$D_i[a_i] = \mathcal{G}_i(2a_i\rho a_i^\dagger - a_i^\dagger a_i\rho - \rho a_i^\dagger a_i) \ , \quad (12)$$

where $\mathcal{G}_1 = \Gamma$, $\mathcal{G}_2 = \mu_1$, $\mathcal{G}_3 = \mu_2$.

Note that the couplings $\lambda_2$ and $\lambda_3$ between the pseudomodes associated with the Matsubara terms and the system are complex (since $c_1$ and $c_2$ are required to be negative), and thus the above Hamiltonian is strangely non-Hermitian[66]. This situation is not immediately covered by the general proof in ref. [43]. We extend that proof in Supplementary Note 6, and show that, to properly take into account the negative $c_1$ and $c_2$, the dynamics of the system has to be computed by solving the following equation of motion for the density matrix $\rho$ (which, throughout this article, will be referred to as the pseudo-Shrödinger equation for simplicity)

$$\frac{d}{dt}\rho = -i[H_{pm}, \rho] + D[\rho] \ , \quad (13)$$

where $D[\rho] = \sum_{i=1}^{3}D_i[a_i]$. The adjective "pseudo" not only refers to the pseudomodes in question but also to the fact that, when $H_{pm}$ is non-Hermitian, we are purposely not taking the Hermitian conjugate when $H_{pm}$ acts on the right of $\rho$.

While we refer to Supplementary Note 6 for a detailed justification, given the non-Hermitian nature of the Hamiltonian in Eq. (11), it is worth presenting here a sketch of the proof.

Following a parallel strategy to the one presented in ref. [43], it is possible to show that the dynamics of observables in the system +pseudomodes space (obtained by solving the pseudo-Shrödinger equation above), is equivalent to a reduced pseudo-unitary dynamics, in which each pseudomode is coupled to a bosonic environment under a rotating-wave approximation and with a constant spectral density (defined for both positive and negative frequencies).

As mentioned, the prefix pseudo- refers to the fact that the Hermitian conjugate is never taken when considering equation of motion for the density matrix. From this auxiliary model, the reduced system's dynamics can be obtained through a Dyson equation. When the pseudomodes and their environments are in an initial Gaussian state, this equation is fully specified by the two-time correlation function of the coupling operator $\sum_{i=1}^{3}\lambda_i(a_i + a_i^\dagger)$.

The advantage of considering a non-Hermitian Hamiltonian together with a pseudo-Schrödinger equation in this derivation is that, by doing so, the Dyson equation for the reduced dynamics of the system is formally equivalent to one where the system is physically interacting with a single environment via a Hermitian coupling operator characterized by the same correlation function $C_0(t) + M_{biexp}(t)$. This completes the proof.

To summarize, the reduced system dynamics computed from Eq. (13) is equivalent to that of the original spin-boson model, Eq. (1), under the assumption (or, in our case, approximation, due to the fitting procedure used to capture the Matsubara terms) that the correlation in Eq. (2) has the form

$$C(t) = C_0(t) + M_{biexp}(t) \ . \quad (14)$$

Remarkably, we will see in a later section that this pseudomode model precisely reproduces the results of the HEOM model, both when the Matsubara frequencies (modes) are neglected, and when they are included, and it also allows for an interpretation of the auxiliary density matrices in the HEOM. In addition, the latter suggests that the HEOM can be derived, in some cases, from the pseudomode model itself (akin to the dissipaton model introduced by Yan[67]). It is also interesting to note that like the HEOM[68], and unlike a normal Lindblad master equation, Eq. (13) does not guarantee complete positivity because of the non-Hermitian couplings. In Supplementary Note 8, we discuss this in detail, and provide criteria for guaranteeing complete positivity in terms of the parameters in the fit.

We finish this section with a brief note on the effect of neglecting the Matsubara correlations, i.e., in considering the approximation $C(t) \mapsto C_0(t)$. In this case, only a single pseudomode is needed, i.e., $i = 1$ in Eqs. (11) and (12). Alternatively, as we show in Supplementary Note 7, this single pseudomode can be understood as mediating the interaction between the system and a residual bath of bosonic modes (with annihilation operator $f_k$ and frequency $\omega_k'$) with the Hamiltonian

$$\begin{aligned}H_{Mats} = &\frac{\omega_q}{2}\sigma_z + \frac{\Delta}{2}\sigma_x + \lambda\sigma_z\frac{(a_1+a_1^\dagger)}{\sqrt{2\Omega}} + \Omega a_1^\dagger a_1 \\ &+ \sum_k \omega_k' f_k^\dagger f_k + \sum_k \frac{g_k'}{\sqrt{2\Omega}\sqrt{2\omega_k'}}\Big(f_k^\dagger a_1 + a_1^\dagger f_k\Big) \ ,\end{aligned} \quad (15)$$

where the couplings $g_\alpha'$ describing the interaction with the residual environment are characterized by the spectral density $J_{Mats}(\omega) = \gamma\Omega$ and defined for both positive and negative frequencies. This system has an interesting relation to another technique used to model the spin-boson model: the reaction coordinate mapping.

**Reaction coordinate (RC) mapping.** Returning to the full spin-boson Hamiltonian, in the reaction coordinate approach a unitary transformation maps the environment to a single-mode reaction coordinate and a residual bath. As discussed in[10,54,69], for the under-damped Brownian motion spectral density the new Hamiltonian is

$$\begin{aligned}H_{RC} = &\frac{\omega_q}{2}\sigma_z + \frac{\Delta}{2}\sigma_x + \lambda\sigma_z\frac{(a+a^\dagger)}{\sqrt{2\omega_0}} + \omega_0 a^\dagger a \\ &+ \sum_k \omega_k'' d_k^\dagger d_k + (a+a^\dagger)\sum_k \frac{g_k''\big(d_k+d_k^\dagger\big)}{\sqrt{2\omega_0}\sqrt{2\omega_k''}} \ ,\end{aligned} \quad (16)$$

where the residual bath, described by operators $d_k$, with frequencies $\omega_k''$ and couplings $g_k''$, has an Ohmic spectral density $J_{res}(\omega) = \gamma\omega$. Importantly, this Hamiltonian is still "exact", and properties of the RC mode are related to the original environment[54] via

$$\sum_k \frac{g_k}{\sqrt{2\omega_k}}(b_k^\dagger + b_k) \equiv \frac{g}{\sqrt{2\omega_0}}(a^\dagger + a) \ . \quad (17)$$

Using this new degree of freedom, for small $\gamma$, such that a Born–Markov-secular approximation for the residual bath is valid, one can derive a new master equation which describes the dynamics of the system coupled to the reaction coordinate, and which preserves detailed balance by definition [see Supplementary Eq. (3) in Supplementary Note 1].

As can be seen by direct comparison, the Hamiltonians in Eqs. (15) and (16) are related by a rotating-wave approximation (RWA) and a Markov approximation in the coupling between the reaction coordinate and its residual environment (and a renormalization of the RC frequency). In Supplementary Note 1, we present an alternative intuitive argument showing why applying a RWA and Markov approximation leads to a correlation function without Matsubara terms. Subsequently, deriving a master equation for the residual environment under these conditions leads to one which does not conserve detailed balance as explicitly shown in Supplementary Eq. (7).

Overall this suggests that the Matsubara frequencies play two roles: first of all, they restore detailed balance, both on the level of the system (in the weak coupling regime, as expected), and also on the level of the system and RC mode (in the strong coupling and narrow-bath regime). Secondly, beyond the weak coupling and narrow-bath regime, they describe the non-negligible influence of "background" modes in the environment not captured by the reaction coordinate itself (e.g., strong correlations with the residual bath).

**Virtual excitations in the ground state.** Before discussing the strong coupling limit, we first consider the weak coupling case and illustrate how neglecting the Matsubara terms leads to an artificial temperature. In Fig. 2, we plot the probability for the qubit to be excited in the steady state $\rho_{11}(t) = \langle 1|\rho(t \to \infty)|1\rangle$ (where $|1\rangle$ is the excited state of the free qubit Hamiltonian) as a function of qubit frequency $\bar{\omega}$. We immediately see that both HEOM and pseudomode approaches give a steady-state population identical to that suggested by Eq. (7) when the Matsubara terms are neglected. Similarly, by fitting the Matsubara terms, and introducing them into both methods, we find that the population drops close to zero, as expected. The residual error in the fit produces a deviation from the expected $\beta = \infty \leftrightarrow \rho_{11}(t \to \infty) = 0$, which becomes large as the qubit frequency becomes small, and hence more sensitive to the residual effective temperature.

As we increase the coupling, the detailed balance condition in terms of the bare qubit Hamiltonian and the original bath temperature in Eq. (7) is no longer a good measure of the fit. This is because in the ultra-strong coupling regime, when the effect of the environment on the system is non-perturbative, so-called virtual excitations can become important[25]. For example, in this scenario, the hybridized system–environment ground state (which in principle should be the steady-state at zero temperature) contains a finite population of photons which cannot be directly observed (or emitted into other modes or environments).

In our treatment of the ultra-strong coupling regime, we find that the Matsubara terms are crucial to obtain the correct photon population in a single collective mode, and trap that population. In order to show this, we first consider the RC picture where the collective bath coordinates are defined in terms of a single mode[54], as per Eq. (17). The RC mapping gives a very clear picture of the dominant influence of the environment in terms of the collective RC mode such that any virtual or real photon population of the collective mode is given by the expectation of the number operator $\langle a^\dagger a \rangle$ (though this does not directly correspond to the original bath-mode occupation).

Can a similar quantity be extracted from the HEOM? It has been shown[70,71] that higher-order moments of the total bath coupling operator can be extracted from certain combinations of auxiliary density operators returned by the HEOM. Similarly, for a single undamped mode, ref. [72] showed that the population is given by the second-level auxiliary density matrix. In our case, we can extract populations that correspond to precisely those of the pseudomodes (see Supplementary Note 2). For example, the

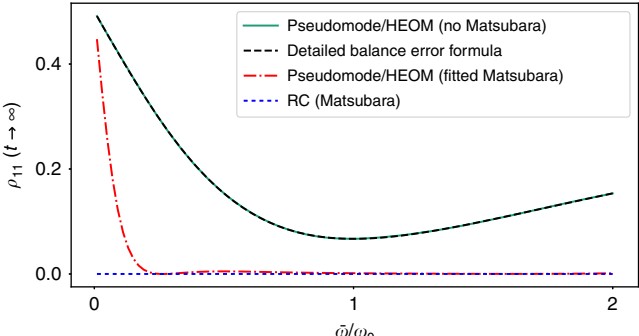

**Fig. 2** Correcting detailed balance by including Matsubara frequencies. The probability of the qubit to be excited in the steady-state limit $\rho_{11}(t \to \infty) = \langle 1|\rho(t \to \infty)|1\rangle$ as a function of the qubit frequency $\bar{\omega}/\omega_0$. Here, we choose a weakly coupled broad bath $\lambda = 0.01\omega_0$, $\gamma = \omega_0$, i.e., when the qubit should have close to zero excitations in the steady state. The black dashed curve is obtained from the effective inverse temperature $\beta_{eff}$ in Eq. (7), and it closely fits the populations of the qubit using both the pseudomode or HEOM methods without Matsubara corrections (turquoise solid curve). This nonzero temperature arises precisely because of the the neglect of the Matsubara terms. The red-dashed-dotted curve shows the results obtained from including the Matsubara terms in the HEOM and pseudomode methods (the results are identical), while the blue double-dashed curve is from the full RC model. We can see that, for the HEOM and pseudomode results, detailed balance is restored up to a residual error from the fit, which, however, can become substantial for small qubit splittings (while the RC model obeys detailed balance by construction). Note that if $\lambda$ is increased, the detailed balance condition in terms of the bare qubit Hamiltonian is not expected to hold in any case, and a more careful error analysis must performed

occupation of the first pseudomode is given by

$$\left\langle a_1^\dagger a_1 \right\rangle = \frac{\rho_{1,1,0,0}}{\lambda_1^2} \quad . \tag{18}$$

It is clear then that the ADOs and the pseudomodes bear a close relationship.

As we can see in Fig. 3 (starting from the initial condition of a zero-temperature environment, and the qubit in the ground state of the free system Hamiltonian), in the absence of the Matsubara terms, the population of the excited state of the two-level system (see Supplementary Fig. 1), and the population of the $a_1$ mode predicted by the HEOM from Eq. (18) match closely that of the RC model with the approximation of the RWA for the RC-residual bath coupling and a flat-residual-bath approximation [described by Supplementary Eq. (7)]. In this case, the population increases to a steady state which can be ascribed to the artificial nonequilibrium situation induced by neglecting the Matsubara correlation. In the RC model without Matsubara contributions, since the state $\rho(t)$ of the qubit and RC mode evolves through the Lindblad equation shown in Supplementary Eq. (7), the rate of energy dissipation into the residual environment is given, in terms of the bare annihilation operator $a$, by

$$J(t) = \gamma\omega_0 \operatorname{Tr}\left[a^\dagger a \rho(t)\right] \quad , \tag{19}$$

i.e., proportional to the average photons in the steady state. However, we know that this emission is unphysical, as it both violates detailed balance and energy conservation.

In contrast, the addition of the Matsubara terms to the HEOM, the addition of the Matsubara modes to the pseudomode model, and the corresponding removal of the unphysical assumptions in the RC model, results in dynamics in all three cases which tend toward a steady state which is close to the ground state of the

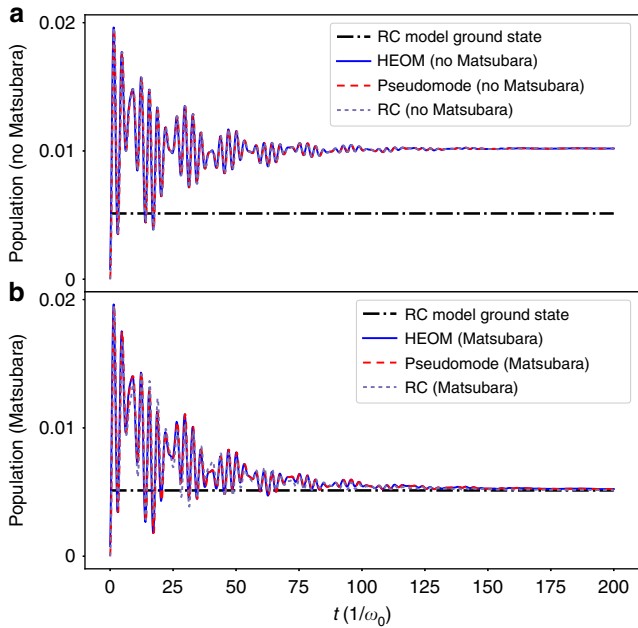

**Fig. 3** Dynamics of the bath-mode occupation for intermediate coupling and narrow bath. For the RC method, we define the bath-mode occupation in terms of the RC mode itself $\langle a^\dagger a \rangle$ (light-purple dashed curves). For the HEOM (blue solid curves) and the pseudomode methods (red-dashed curves), the bath mode is the effective mode associated with the frequency $\Omega$. The parameters are $\lambda = 0.2\omega_0$, $\gamma = 0.05\omega_0$, $\omega_q = 0$, $\Delta = \omega_0$, $T = 0$. The upper panel (**a**) gives the results of the three models we consider without Matsubara terms (both direct, and effective in the RC case). For this choice of parameters, all three models coincide. In the lower panel (**b**), we show the three models with Matsubara terms included, and all three tend towards a steady-state which corresponds to the ground state of $H_{RC}$ (dashed-dotted black line)

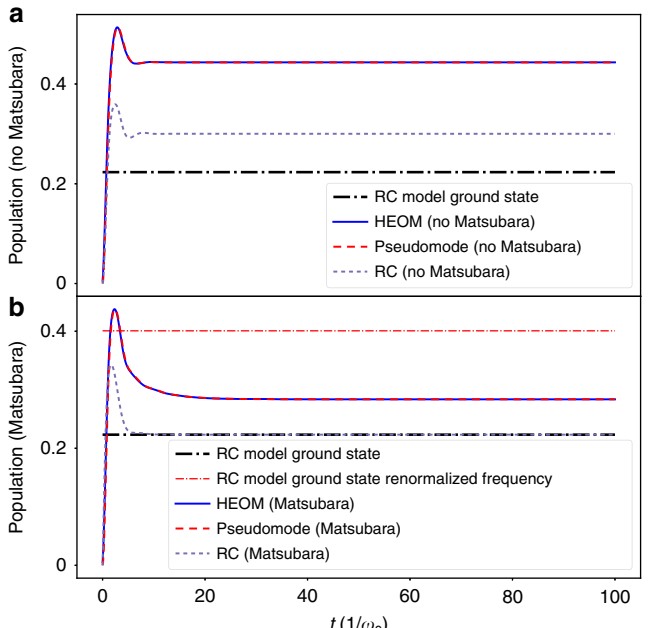

**Fig. 4** Dynamics of the bath-mode occupation for ultra-strong coupling and broad bath. Here, we set the parameters $\lambda = \omega_0$, $\gamma = \omega_0$, and again $\omega_q = 0$, $\Delta = \omega_0$, $T = 0$. The upper panel (**a**) gives the results of the three models we consider without Matsubara terms (both direct, and effective in the RC case). For this choice of parameters, all the HEOM (blue solid curve) and pseudomode models (red-dashed curve) coincide, but the RC model (light-purple dashed curve) shows some deviations because it does not take into account the renormalized frequency $\Omega$. In the lower panel (**b**), we show the three models with Matsubara terms included, and now only the RC model tends to the ground state of $H_{RC}$, while the pseudomode and HEOM models coincide and take into account corrections due to strong correlations with the effective "Matsubara modes" (note that the RC model is not corrected by just including the renormalized frequency, as shown by the red dot-dashed line, which shows the ground-state occupation for an RC model with a phenomenologically altered frequency, i.e., by setting the frequency of the RC mode to be equal to $\Omega$)

coupled system-RC Hamiltonian. In this case, the HEOM and pseudomode model match exactly, while the RC model gives a qualitative agreement. This trend is one of our primary results: the addition of Matsubara terms to the HEOM (or equivalently Matsubara modes to the pseudomode model) restores detailed balance in terms of the coupled system-RC Hamiltonian (up to a residual error from the fit) and traps photons in an effective ground state, as confirmed by the RC model. In this case, the state $\rho(t)$ of the qubit and RC mode evolves through the Lindblad equation shown in Supplementary Eqs. (1) and (3) characterized by jump operators between eigenstates. As a consequence, since the steady state is the ground state, there is no steady-state energy dissipation [see Supplementary Eq. (4)] into the residual bath.

As $\gamma$ is increased, cf. Fig. 4, we see a deviation between HEOM and RC models (see also Supplementary Fig. 1 for a comparison of system populations, and Supplementary Note 5 for a discussion of the steady state as a function of coupling strength). For strong coupling and broad baths, the Matsubara terms become more relevant, as does the error arising from the fitting procedure. In Supplementary Note 4, we perform an error analysis which suggests that the difference between the RC results and the HEOM results exceed potential errors arising from the fit. Thus, we primarily ascribe this difference to the breakdown of the perturbative approximation for the residual bath in the RC model, which becomes more pronounced as $\gamma$ is increased.

One might attribute the difference to the fact that the RC model does not take into account the frequency shift that we see in Eq. (4). However, phenomenologically solving for the ground state of the system coupled to an RC mode with renormalized

frequency $\Omega$ actually predicts a larger population (shown by the red dot-dashed line in Fig. 4) than the normal system-RC ground state due to the decreased frequency of the non-Matsubara mode[25]. In addition, the predicted population is also larger than the full HEOM/pseudomode results, which suggests that, as $\gamma$ is increased, the correlations between the system and the pseudomodes associated with the Matsubara frequencies become stronger, and actually reduce the population in the non-Matsubara pseudomode[25]. However, without the RC model to guide us with a physical interpretation in this limit, it becomes difficult to associate the populations of the Matsubara modes to real physical modes, collective or otherwise[53,73–76]. In fact, as described earlier, since their contribution to the correlation functions of the bath is negative in the parameter regimes we consider here, in the pseudomode model their coupling to the system is non-Hermitian, accentuating their nature as simulacra.

## Discussion
We have analyzed the dynamics and steady-state properties of the zero-temperature spin-boson model in the strong and ultra-strong coupling regime using three different techniques. We showed that the Matsubara terms, taken into account with a fitting procedure in the HEOM and pseudomode methods, restore detailed balance (albeit up to a residual error), even in the ultra-

strong coupling regime. This was validated by a comparison with the reaction coordinate method, which also indicates that the Matsubara terms are important for the correct "trapping" of virtual excitations in the collective ground state.

Simultaneously, we showed that a pseudomode model can exactly capture the same dynamics as the HEOM, and can take into account negative contributions to the correlation functions, like the Matsubara frequencies, via a pseudo-Schrödinger equation.

Our results also elucidate the relationships and differences between the three methods employed herein, particularly the strong relationship between the pseudomode treatment and the HEOM.

Future work includes generalizing to arbitrary spectral densities for systems such as superconducting qubits coupled to transmission lines (with potentially structured environments[77]), and photosynthetic complexes[4–9]. In addition, in the broad-bath limit, it may be possible to assign direct physical meaning to the ADOs of the HEOM, and the Matsubara modes of the pseudo-mode method, by comparison with bosonic-chain mappings of the environment[53,73–76,78], in the same way the RC mapping guides us in this work. This might allow, for example, inspection of spatial dependencies of the photon population, as revealed by other methods[32,33]. We hope that these insights can help toward a better understanding of ultra-strong coupling at zero temperature in continuum systems, and emphasize the impact of the positive frequency nature of many physical environments (and the resulting appearance of Matsubara frequencies).

## Data availability

Dataset sharing is not applicable to this article as no data sets were generated or analyzed during this study.

## Code availability

The numerical code used to generate most of the figures in this work is available at https://doi.org/10.5281/zenodo.3294068 and https://github.com/pyquantum/matsubara, and is documented at https://matsubara.readthedocs.io/. It is available to use under the MIT license, and uses the open-source library QuTiP[79,80]

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

## Acknowledgements

We would like to thank Stephen Hughes for helpful suggestions on the pseudomode approach, and Ken Funo, David Zueco, Simone De Liberato, Huan-Yu Ku, and Fabrizio Minganti for feedback and comments. We would also like to thank Fabio Mascherpa for bringing to our attention their recent work[78]. S.A. was supported by the RIKEN IPA program. N.L. acknowledges support from JST PRESTO, Grant no. JPMJPR18GC. N.L. and F.N. acknowledge support from the RIKEN-AIST Joint Research Fund and the Sir John Templeton Foundation. F.N. is partly supported by the MURI Center for Dynamic Magneto-Optics via the Air Force Office of Scientific Research (AFOSR) (FA9550-14-1-0040), Army Research Office (ARO) (Grant no. W911NF-18-1-0358), Asian Office of Aerospace Research and Development (AOARD) (Grant no. FA2386-18-1-4045), Japan Science and Technology Agency (JST) (the Q-LEAP program, and CREST Grant no. JPMJCR1676), Japan Society for the Promotion of Science (JSPS) (JSPSRFBR Grant no. 17-52-50023, JSPS-FWO Grant no. VS.059.18N). M.C. acknowledges support from NSAF no. U1730449.

## Author contributions

N.L. and S.A. contributed equally to the development of the numerical simulations. M.C. performed analytical calculations, and developed the proof of the pseudomode method. N.L. conceived the project. N.L and F.N. supervised the research. All authors discussed the results and contributed to writing the paper.

## Additional information

**Competing interests:** The authors declare no competing interests.

