## [Peer Review File · Nature Communications]

Reviewers' comments:

Reviewer #1 (Remarks to the Author):

In this paper the authors present a general, non-perturbative treatment of the open-system dynamics in the spin-boson model, focusing in particular on the strong-coupling regime and on the related generation of virtual excitations. Starting from the decomposition of the environmental correlation function into, respectively, a "resonant" contribution and a (possibly infinite) sum of exponentials, fixed by the so-called Matsubara frequencies, the authors first provide an intuitive motivation to show how neglecting the Matsubara contributions (totally or in part) would unavoidably lead to a significant violation of basic principles, such as the detailed balance condition. Importantly, (some of) the Matsubara frequencies are often neglected in the description of the dynamics of low temperature systems, for example when using standard HEOM techniques. The authors then propose a general method to take into account properly also the Matsubara contribution to the correlation function. The method is essentially based on a direct fit of the correlation function with exponentials; in the zero-temperature limit two exponentials are enough to reproduce the Matsubara contribution (instead of the standard truncated series of exponentials). Relying on that, it is then possible to introduce a set of pseudo-modes, which are coupled to the spin and which lead to the same open-system dynamics as in the original system. Crucially, the pseudo-modes introduced here evolve via a Lindblad equation whose coherent part is actually a pseudo-Schroedinger equation, since it involves a non-Hermitian Hamiltonian (which is still commuting with the state, see Eq.(13)). Such a representation also allows for a direct comparison with the reaction-coordinate (RC) method [see Eqs.(14) and (15)], leading to the conclusion that neglecting the Matsubara terms corresponds to both a rotating wave and a Markov approximation in the RC method. The paper then deals with the appearance of virtual photons in the long-time dynamics of a strongly coupled system, a phenomenon which essentially consists in a finite photon population due to the relaxation toward a hybridized system-environment "ground state". The authors show that neglecting the Matsubara contributions would lead to an unphysical prediction, overestimating the spin and bath pseudomode asymptotic populations [see figures 2(a) 4(a) and 5(a)], while the inclusion of the Matsubara terms ensures to the expected result. The predictions obtained via the pseudomode approach match those obtained via the HEOM method, while there is a deviation with respect to the RC predictions, especially for a very strong coupling; indeed, such a deviation is due to the approximated treatment of the residual bath in the RC method [see also Appendix A].

The description of general dynamics of open quantum systems beyond the weak coupling and/or Markovian regime is at the same time a challenging and very interesting topic, with potential applications to the characterization of several different physical systems, in which one wants to fully capture the complexity of the environmental effects on the open system. The approach presented here, which relies on an extension of the pseudomode techniques based on the use of a non-Hermitian Hamiltonian, is general and flexible enough to represent a promising way to deal with strongly coupled systems and non-Markovian dynamics. This is clearly shown by the analysis on the ground-state virtual excitations induced by the dynamics on the long time, where the present approach, allowing to treat in a more complete way the Matsubara contributions, guarantees the correct dynamical features up to the steady state. In addition, let me also emphasize that the analysis of the authors clarifies the connection between different methods which are routinely used to describe (non-Markovian) open-system dynamics (pseudomodes, RC and HEOM), which is certainly a relevant added value to the work.

My assessment of the paper is thus certainly positive. Before fully recommending it for publication in Nature Communication, I would like the authors to address the following points.

- My main concern is related to the statement, which is repeated several times (including in the abstract, introduction and conclusion), that the present analysis restores the validity of the detailed balance condition. Now, while it is clear that removing the Matsubara contributions leads

to a violation of the detailed balance, it appears not obvious to me that the latter is fully reestablished by the inclusion of the Matsubara contribution via the exponential fit expressed in Eq.(9). Such a fit will unavoidably bring along an error (as extensively investigated in the Appendix), which might manifest in a violation of the detailed balance condition; in fact, there is a priori no guarantee that the pseudo-Lindblad form in Eq.(13) will satisfy the detailed balance condition (as expressed, for example, in Eq.(6)). I think that the authors should explain this point more in detail, or show explicitly that the detailed balance condition is indeed satisfied.

- I find the discussion at the end of the first column and beginning of the second column at page 6 not completely clear: on the one hand, it is shown that the correlation function of the original bath reduces to the single non-Matsubara term when the RW and Markov approximations (the latter with respect to the residual bath) are performed; on the other hand, it is stated that the mentioned analysis connects Eq.(15) and (16). Can't one just argue that applying the RWA and the Markov approximation (in the sense of taking a flat spectrum) to Eq.(16) one precisely obtains Eq.(15)? Wouldn't this be equivalent to what has been shown with respect to the correlation function or one does need separately both analyses?

- At page 4 in the first column, it is argued that the non-Matsubara part can be reduced to the sum of 2 exponents, but then, in the second column of page 4 and for the rest of the paper, one single under-damped mode is considered. Is here an approximation implied?

- Is it correct to say that the Dyson expansion generated by Eq.(F8) still takes the form as in Eq.(F2), despite the non-Hermiticity of the Hamiltonian, because such non-Hermiticity is present only in the S-B' interacting term and not in the free terms (otherwise the whole interaction picture would have to be changed)? If this is the case, it might be worth to point it out explicitly.

- Please, note that the same symbol ω_k is used with two different meanings in Eq.(1) and Eq.(5); for the sake of clarity, it might be convenient to change one of the two. Moreover, there is a typo in \mathcal{G}_i (which should read \mathcal{G}_3) after Eq.(12) and the inset mentioned in the caption of Fig.6 seems to be missing.

Reviewer #2 (Remarks to the Author):

The authors address the problem of treating virtual excitations in the regime of ultra-strong coupling by the spin-boson model. This has been an open issue, due to the difficulty to explore these features in systems with a continuous environment. In fact, such a study requires non-perturbative and non-Markovian procedures, for instance by using hierarchical equations of motion (HEOM). The HEOM however remain unsuitable to treat low-temperature regimes, linked to the difficulty of analytically decomposing the bath correlation function into a finite sum of exponentials. Albeit the possibility of fitting the total correlation-function to exponentials for use with the HEOM has been previously explored (see Refs. [48,50,53]), the results did not provide useful physical insights because of the fitting error. Here the authors go further thanks to an approach which separates the correlation function into an analytical part, comprised of the desired finite number of exponentials, and an additional part (made of Matsubara modes) given by an infinite sum of exponentials. This approach eventually supply physical results and interpretations starting from apparently "unphysical (Matsubara) modes". In fact, they find that neglecting the Matsubara part in the bath correlation function leads to unphysical consequences, such as emission of energy from a collective ground-state in the reaction coordinate model. Moreover, they find that including Matsubara modes for the usual pseudo-mode model with non-Hermitian coupling to the system can reproduce the HEOM results for all ranges of system parameters.

I find the manuscript of interest and technically sound. The results give useful insights for the interpretation and understanding of physical phenomenons of open quantum systems (described

by the spin-boson model) under ultra-strong coupling conditions. However, I believe that the manuscript requires a revision to comply with the requirements of the journal regarding novelty and impact in the wider field. I list in the following my comments.

1) Abstract. I find the abstract full of details and uneasy to be read for a non-specialist. The concept of Matsubara modes is not of general knowledge and, seeing the general nature of the journal, I believe that a brief explanation of their role would help the reader. Moreover, the last sentence of the abstract, where the presence of a non-Hermitian pseudo-Schrodinger equation is taken to justify the fact that apparently unphysical modes can give rise to physical behaviour cannot be clear for a non-specialist. The fact itself that non-Hermitian coupling or Hamiltonian can produce real (physical) eigenvalues has been firstly shown in Phys. Rev. Lett. 80, 5243 (1998), which appears not to be cited in the manuscript. In addition, the term pseudo-Schrodinger equation is named here by the authors and is not clear from the abstract how this can clarify the emergence of a physical behaviour as a consequence.

I would suggest the authors to try and structure the abstract with an initial background and rationale, then followed by their main contributions and final considerations. An abstract containing the main points and their implications will be very helpful to a broader audience.

2) My general point about the nature of the manuscript is with its current presentation for a broad readership. The Introduction contains many technical details, especially in page 2, and requires a revision in order to be clear about both the state-of-art and the original contribution of the authors going beyond the state-of-art. I suggest to simplify the Introduction so to be more direct and focusing on the main aspects. Some of the following questions should be principally addressed:

- Why is the problem of virtual excitations in the ultra-strong coupling regime important from a general perspective?
- Which is their role in different physical contexts (the authors give some indications at the beginning of the right column of page 1)?
- What does it mean "discrete bosonic mode" and then "continuum systems (the authors should clarify that the continuum system is indeed the environment made of many bosonic modes (harmonic oscillators), the system of interest being always a qubit (spin)))?
- Which is the original approach adopted here by the authors allowing one to overcome the problems in treating virtual excitations under general conditions of the open system?

In general, I believe that the authors should make an effort to make their abstract and introduction of broader view, being compact and clear also to a non-specialist.

Reviewer #3 (Remarks to the Author):

The authors study the effect of the environment in the seminal spin-boson model, mainly focusing on the zero-temperature limit and including the ultra-strong coupling regime, by developing a method that goes beyond the usual perturbative approaches and assumptions (Born-Markov and RWA). It is known that the effect of a bath on a given systems is captured by the correlation function of the full bath. The authors focus on the under-damped limit of the spectral density, which is the regime relevant to a wide variety of physical systems. In this limit, they find that the correlation function (the quantity relevant to the system dynamics) can be decomposed into an analytical part, that represent the resonant part of the part and can be fitted to two exponential at low-temperature, and an infinite sum of exponential, that they define as 'Matsubara part' because of its dependence on the Matsubara frequencies.

As a key result, they show that the continuum is important to model non-Markovian dynamics, and to respect detailed balance — necessary to model the correct equilibrium properties.

Their approach goes beyond currently known methods, and they show compelling comparisons that support this argument: (i) In the zero-temperature limit, they extend the HEOM method by representing the Matsubara part as an integral, that is further accurately fitted to a bi-exponential.

This largely simplifies computation; (ii) They compare their results to the pseudo-modes, that they further extend to allow for complex coupling values (non-Hermitian Hamiltonian), (iii) and include a comparison with results obtained from reaction coordinate mapping. Their analysis extend to the ultra-strong coupling regime, where it is shown that the Matsubara terms are crucial to get the correct populations.

The authors nicely state the limitations of their model and its range of validity.

In short, the authors provide an in-depth analysis of how to simulate coupling to an under-damped environment, presenting a new decomposition of the spectral density, as well as extending other known methods. Their work is nicely integrated in comparison with other known methods. The manuscript will be useful for experts in the field of quantum dynamics, and might well influence future research in the simulation of more complex systems. Therefore, I recommend this work for publication in Nature Communication.

I have only very few suggestions that the authors might consider to improve their work, which is already nicely presented:

1. The fact that the Matsubara correlations can be described by a bi-exponential at low temperature is interesting. Do the authors have any intuition of the physics captured by the relaxation times μ_1 and μ_2 in Eq. 9?

2. In the pseudo-mode treatment, the system is coupled to modes that are dissipative (with Lindblad dissipation). Yet the authors find that the coupling constant λ_1 and λ_2 become complex. Does this suggest an additional form of decay for the modes directly coupled to the system, that is not accounted for by the Lindblad decay, as in e.g. Ref. [59]? In that case, I wouldn't qualify the Hamiltonian as 'purely mathematical' (last sentence before the conclusion), but rather say that the microscopic nature of the decay of the pseudo-modes is not purely Lindbladian.

3. Add reference to justify Eq. 19, or an explanatory paragraph as to why a single mode is enough in the RC mapping.

Aurelia Chenu

We wish to thank all referees for the positive criticism and feedback. We have included responses to all questions and suggestions below. Corresponding changes the main article and supplementary information are marked in red therein.

Referee 1

My main concern is related to the statement, which is repeated several times (including in the abstract, introduction and conclusion), that the present analysis restores the validity of the detailed balance condition. Now, while it is clear that removing the Mastrubara contributions leads to a violation of the detailed balance, it appears not obvious to me that the latter is fully reestablished by the inclusion of the Matsubara contribution via the exponential fit expressed in Eq.(9). Such a fit will unavoidably bring along an error (as extensively investigated in the Appendix), which might manifest in a violation of the detailed balance condition; in fact, there is a-priori no guarantee that the pseudo-Lindblad form in Eq.(13) will satisfy the detailed balance condition (as expressed, for example, in Eq.(6)). I think that the authors should explain this point more in detail, or show explicitly that the detailed balance condition is indeed satisfied.

This is an interesting question. Essentially yes, the Referee is correct in that, with the fitting approach, the full exact detailed balance should not be expected to be completely restored as long as we have a fit with some error.

In our manuscript we have used Eq. (7) primarily as a way to quantify the breaking of the detailed balance when the coupling to the environment is small and when the Matsubara contributions are neglected completely. Once the fit is done, and the approximate Matsubara modes included in the HEOM or Pseudo-mode models, one can in principle still use detailed balance to verify the quality of the fit, albeit only in the limit of weak coupling to the environment (since at strong coupling we no longer expect equation (7) to hold).

Since the system is a qubit, this would mean that, because of the imperfect fit, the observed temperature of the qubit differs from the temperature of the bath. We investigated this to some degree in the appendix, on page 16 in the previous version, when discussing the small coupling limit of figure 8. There one can precisely see that in that limit there is a residual very small 'effective temperature' due to the error in the fit. We have clarified this at various points in the main text, by accentuating the point that the restoration of detailed balance still suffers from the error in the fit. In addition, we have substantially modified what was figure 3 (*now figure 2, we changed the order of the figures to improve flow of the results section*) to give an explicit example, and also added a more detailed discussion of that figure in the text in on pages 6 and 7. In addition, in that figure, instead of plotting the inverse temperature we explicitly show the excited state probability of the qubit itself.

However, when the coupling with the environment is strong (and even more so when it is both strong and broad), the detailed balance given by Eq. (7) is no longer a good measure of the fit. Comparison with the RC method suggests that for narrow baths detailed balance in terms of the system+RC mode should hold (i.e., we should get a steady-state which looks like the ground state of that model, as we show in what is now figure 3 (was figure 2)). For broader baths this also breaks down, so we must perform a more general error analysis, as shown in the appendix.

Referee 1

I find the discussion at the end of the first column and beginning of the second column at page 6 not completely clear: on the one hand, it is shown that the correlation function of the original bath reduces to the single non-Matsubara term when the RW and Markov approximations (the latter with respect to the residual bath) are performed; on the other hand, it is stated that the mentioned analysis connects Eq.(15) and (16). Can't one just argue that applying the RWA and the Markov approximation (in the sense of taking a flat spectrum) to Eq.(16) one precisely obtains Eq.(15)? Wouldn't this be equivalent to what has been shown with respect to the correlation function or one does need separately both analyses?

Yes, the referee is correct; in principle one can directly apply RWA and Markov approximations to equation (16)'s residual bath interaction, and make the same point. (note however that the frequency of the RC mode will still differ from that of the PM in equation 15).

Our motivation for the discussion following equation 16 was to give an alternative intuitive description of the same ideas. We have clarified this in the text, and more explicitly pointed out the relation between equations 15 and 16. We have also moved the majority of this intuitive argument to the supplementary material, to make the logic easier to follow.

Referee 1

At page 4 in the first column, it is argued that the non-Matsubara part can be reduced to the sum of 2 exponents, but then, in the second column of page 4 and for the rest of the paper, one single under-damped mode is considered. Is here an approximation implied?

There is no approximation at that stage, this is a just consequence of how the parameters are included in the HEOM method. In reality, at zero temperature, equation (4) is just one exponent, and hence is described by just one damped pseudomode. In the HEOM, the real and imaginary parts of the correlation function are parameterized separately, hence the somewhat convoluted decomposition on page 4. We have clarified this in the text, and thank the referee for pointing out that this was unclear. (Related to this point, we also clarified the caption of figure 1).

Referee 1

Is it correct to say that the Dyson expansion generated by Eq.(F8) still takes the form as in Eq.(F2), despite the non-Hermiticity of the Hamiltonian, because such non-Hermiticity is present only in the S-B' interacting term and not in the free terms (otherwise the whole interaction picture would have to be changed)? If this is the case, it might be worth to point it out explicitly.

Yes, the Referee is correct. The Dyson equation that Eq. (F8) (now Eq. (27)) would generate should be formally equivalent to the one in (F2) (now Eq. 21). However, one has now the freedom to choose both the interaction terms and the free Hamiltonian of the bath as long as Eq. (F7) (now Eq. 26) is satisfied, and as long as the whole process is Gaussian, which allow the equivalence of the Dyson equations. As noted by the Referee, a change in the free dynamics would require one to modify the interaction picture accordingly (in this case, derived from the pseudo-Schrodinger equation as in (F14) (now Eq. (33)). We have edited the text in this section to clarify this point. We have also fixed a typo in equation F(18) (now Eq. 37). *(Note that the equation numbers have changed as we moved the pseudomode proof into the methods section).*

Referee 1

- Please, note that the same symbol ω_k is used with two different meanings in Eq.(1) and Eq.(5); for the sake of clarity, it might be convenient to change one of the two. Moreover, there is a typo in \mathcal{G}_i (which should read \mathcal{G}_3) after Eq.(12) and the inset mentioned in the caption of Fig.6 seems to be missing.

We thank the referee for spotting these typos/issues. In equation (5), and where appropriate in the text, we now use ϵ_k instead of ω_k to describe the Matsubara frequencies.

Previously the insets in figure 6 were removed for clarity, but we neglected to update the caption. This has been fixed.

Referee 2

1) Abstract. I find the abstract full of details and uneasy to be read for a non-specialist. The concept of Matsubara modes is not of general knowledge and, seeing the general nature of the journal, I believe that a brief explanation of their role would help the reader. Moreover, the last sentence of the abstract, where the presence of a non-Hermitian pseudo-Schrodinger equation is taken to justify the fact that apparently unphysical modes can give rise to physical behaviour cannot be clear for a non-specialist. The fact itself that non-Hermitian coupling or Hamiltonian can produce real (physical) eigenvalues has been firstly shown in Phys. Rev. Lett. 80, 5243 (1998), which appears not to be cited in

the manuscript. In addition, the term pseudo-Schrodinger equation is named here by the authors and is to clear from the abstract how this can clarify the emergence of a physical behaviour as a consequence.

I would suggest the authors to try and structure the abstract with an initial background and rationale, then followed by their main contributions and final considerations. An abstract containing the main points and their implications will be very helpful to a broader audience.

We thank the referee for his feedback and advice. We have now completely rewritten the abstract to be more accessible to a general audience, and more clearly state the rationale of our work.

Referee 2

2) My general point about the nature of the manuscript is with its current presentation for a broad readership. The Introduction contains many technical details, especially in page 2, and requires a revision in order to be clear about both the state-of-art and the original contribution of the authors going beyond the state-of-art. I suggest to simplify the Introduction so to be more direct and focusing on the main aspects. Some of the following questions should be principally addressed:

- Why is the problem of virtual excitations in the ultra-strong coupling regime important from a general perspective?
- Which is their role in different physical contexts (the authors give some indications at the beginning of the right column of page 1)?
- What does it mean "discrete bosonic mode" and then "continuum systems (the authors should clarify that the continuum system is indeed the environment made of many bosonic modes (harmonic oscillators), the system of interest being always a qubit (spin))?
- Which is the original approach adopted here by the authors allowing one to overcome the problems in treating virtual excitations under general conditions of the open system?

In general, I believe that the authors should make an effort to make their abstract and introduction of broader view, being compact and clear also to a non-specialist.

This is also very valuable advice, and we have made an attempt to implement these clarifications in the introduction.

Referee 3

The fact that the Matsubara correlations can be described by a bi-exponential at low temperature is interesting. Do the authors have any intuition of the physics captured by the relaxation times μ_1 and μ_2 in Eq. 9?

The choice of using two exponentials is ultimately a trade-off between keeping the numerical cost of the resulting pseudo-mode or HEOM simulation low and getting a good fit. The Matsubara terms have a polynomial decay so it is necessary to use more than one exponential in order to fit them.

As for an intuition about the physics of the relaxation times, as we describe in the paper we primarily understand them via what happens when they are neglected: detailed balance is lost, and the non-Matsubara part of the environment picks up some unjustified properties.

In the strong coupling case, where the ‘Matsubara modes’ also begin to be correlated with the system, the meaning of these relaxation times from the point of view of the system is less clear, and remains an open question (see below, to the referees related question).

Referee 3

In the pseudo-mode treatment, the system is coupled to modes that are dissipative (with Lindblad dissipation). Yet the authors find that the coupling constant λ_1 and λ_2 become complex. Does this suggest an additional form of decay for the modes directly coupled to the system, that is not accounted for by the Lindblad decay, as in e.g. Ref. [59]? In that case, I wouldn’t qualify the Hamiltonian as ‘purely mathematical’ (last sentence before the conclusion), but rather say that the microscopic nature of the decay of the pseudo-modes is not purely Lindbladian.

Generally speaking we cannot associate the non-hermitian nature of couplings to a “decay”. This makes them differ from non-hermitian frequencies per se, though they may indirectly create additional decay channels because of their overdamped frequencies μ_1 and μ_2 .

In fact, as we argue in the paper, these couplings can describe a kind of extra correlations with the residual environment. For example, an extreme limit we took in trying to understand this problem, but one that didn’t include in the manuscript, is that of a TLS coupled to a single mode (i.e., a loss-less purely Hermitian Rabi model). In that case we can artificially split the single-mode into two, one mode with larger coupling, and one with imaginary coupling, such the total correlation function of the original one-mode and artificial two-mode problems are the same. In comparing these two models (original and artificial), one observes identical dynamics for the TLS. In both examples there is no decay in the model and the extra imaginary coupling just helps in restoring the correct “non-Markovian” reduced dynamics expected for the original ‘single mode problem’.

For the latter part of the referee’s question, we do agree that in the strong coupling case these extra modes do modify the purely Lindblad decay used in reference 59. It is interesting to note that, unlike a purely Lindbladian equation, the pseudo-Schrodinger equation does not guarantee

positivity if the error in the fit is very large. For example, we found that for complete positivity a minimal requirement is that one must always have a total free-bath correlation function that corresponds to a physical (positive) power spectrum. We have added a short section on this in the appendix, and briefly discuss it on page 5. We appreciate the referee for pointing this out, and motivating us to include this additional result.

Our use of the word 'purely mathematical' was more of a reference to the simulacra nature of the pseudomodes, in the sense that they are designed to mimic the influence of the true original environment on the system. We have modified this sentence to make this clearer.

Referee 3

Add reference to justify Eq. 19, or an explanatory paragraph as to why a single mode is enough in the RC mapping.

We thank the referee for pointing out this lack of definition/justification. To clarify this we have moved equation (19) to directly after equation (16), and in the text we have clarified that the coupling to the environment is through the operator written on the left side of what was Eq. (19), and the RHS is defined by a simple transformation; this corresponds to the quadrature of a newly defined harmonic oscillator (the RC). We have also clarified that equation (16) and what was (19) are 'exact' but that the overall RC model is approximate (at the level of the approximations made to include the 'residual bath' as a master equation). As we show in the results section, this approximation can break down for large coupling to that residual bath.

Finally, we thank all three referees for their constructive comments, which have helped us to significantly improve the presentation and clarity of our work. We are very grateful for their feedback and insights.

Additional changes:

- 1) We have moved the proof of the pseudomode method from the supplementary information into the 'Methods' section.
- 2) We have formatted the layout (section titles) to fit the requirements of Nature Comms.
- 3) We have improved the readability of the figures.

REVIEWERS' COMMENTS:

Reviewer #1 (Remarks to the Author):

I thank the authors for their reply and for fully addressing the issues I made in my previous report. As a side note, I think that also the other new parts of the updated version of the manuscript, especially the discussion about the complete positivity of the extended pseudomode approach, add further value to the analysis.

Hence, I do recommend the paper for publication in Nature Communications.

Reviewer #3 (Remarks to the Author):

The authors have addressed all my comments. The manuscript has been much improved with the revision. It seems adequate for publication in Nature Communication in the current form.